# Continuous Deep Equilibrium Models: Training Neural Odes Faster by Integrating Them to Infinity

## Abstract

Implicit models separate the definition of a layer from the description of its solution process. While implicit layers allow features such as depth to adapt to new scenarios and inputs automatically, this adaptivity makes its computational expense challenging to predict. In this manuscript, we *increase the "implicitness" of the DEQ by redefining the method in terms of an infinite time neural ODE*, which paradoxically decreases the training cost over a standard neural ODE by $2 - 4 \times$. Additionally, we address the question: *is there a way to simultaneously achieve the robustness of implicit layers while allowing the reduced computational expense of an explicit layer?* To solve this, we develop Skip and Skip Reg. DEQ, an implicit-explicit (IMEX) layer that simultaneously trains an explicit prediction followed by an implicit correction. We show that training this explicit predictor is free and even decreases the training time by $1.11 - 3.19 \times$. Together, this manuscript shows how bridging the dichotomy of implicit and explicit deep learning can combine the advantages of both techniques.

## 1 Introduction

Implicit layer methods, such as Neural ODEs and Deep Equilibrium models Chen et al. (2018); Bai et al. (2019); Ghaoui et al. (2020), have gained popularity due to their ability to automatically adapt model depth based on the "complexity" of new problems and inputs. The forward pass of these methods involves solving steady-state problems, convex optimization problems, differential equations, etc., all defined by neural networks, which can be expensive. However, training these more generalized models has empirically been shown to take significantly more time than traditional explicit models such as recurrent neural networks and transformers. *Nothing within the problem's structure requires expensive training methods, so we asked, can we reformulate continuous implicit models so that this is not the case?*

Grathwohl et al. (2018); Dupont et al. (2019); Kelly et al. (2020); Finlay et al. (2020) have identified several problems with training implicit networks. These models grow in complexity as training progresses, and a single forward pass can take over 100 iterations (Kelly et al., 2020) even for simple problems like MNIST. Deep Equilibrium Models (Bai et al., 2019; 2020) have better scaling in the backward pass but are still bottlenecked by slow steady-state convergence. Bai et al. (2021b) quantified several convergence and stability problems with DEQs. They proposed a regularization technique by exploiting the "implicitness" of DEQs to stabilize their training. *We marry the idea of faster backward pass for DEQs and continuous modeling from Neural ODEs to create Infinite Time Neural ODEs which scale significantly better in the backward pass and drastically reduce the training time.*

Our main contributions include[1]

1. An improved DEQ architecture (Skip-DEQ) that uses an additional neural network to predict better initial conditions.

---

[1]We provide an anonymous version of our codebase `https://anonymous.4open.science/r/continuous_deqs_infinite_time_neural_odes/` with the intent of public release after the review period

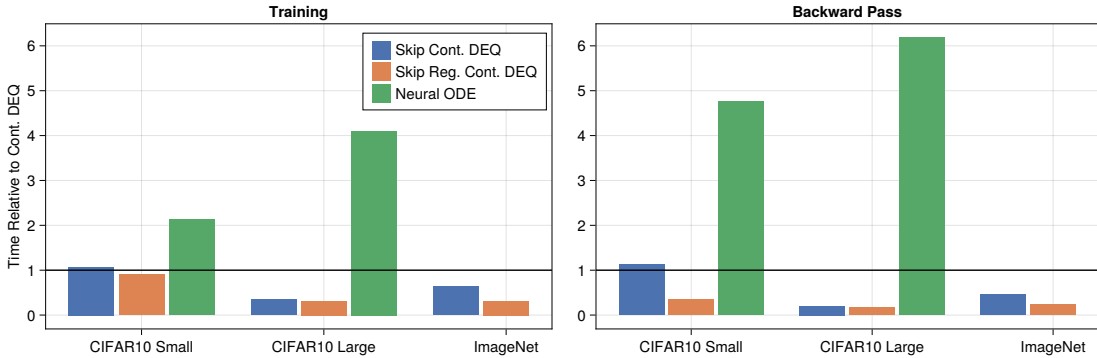

Figure 1: **Relative Training and Backward Pass Timings against Continuous DEQs** *(lower is better)*: In all scenarios, Neural ODEs take *4.7 − 6.182×* more time in the backward pass compared to Vanilla Continuous DEQs. Whereas combining Skip (Reg.) with Continuous DEQs accelerates the backward pass by *2.8 − 5.9×*.

2. A regularization scheme (Skip Regularized DEQ) incentivizes the DEQ to learn simpler dynamics and leads to faster training and prediction. Notably, this does not require nested automatic differentiation and thus is considerably less computationally expensive than other published techniques.

3. A continuous formulation for DEQs as an infinite time neural ODE, which paradoxically accelerates the backward pass over standard neural ODEs by replacing the continuous adjoints with a simple linear system.

4. We demonstrate the seamless combination of Continuous DEQs with Skip DEQs to create a drop-in replacement for Neural ODEs without incurring a high training cost.

## 2  Background

Explicit Deep Learning Architectures specify a projection $f : \mathcal{X} \mapsto \mathcal{Z}$ by stacking multiple "layers". Implicit models, however, define a solution process instead of directly specifying the projection. These methods enforce a constraint on the output space $\mathcal{Z}$ by learning $g : \mathcal{X} \times \mathcal{Z} \mapsto \mathbb{R}^n$. By specifying a solution process, implicit models can effectively vary features like depth to adapt automatically to new scenarios and inputs. Some prominent implicit models include Neural ODEs (Chen et al., 2018), where the output $z$ is defined by the ODE $\frac{dz}{dt} = g_\phi(x, t)$. Liu et al. (2019) generalized this framework to Stochastic Differential Equations (SDEs) by stochastic noise injection, which regularizes the training of Neural ODEs, allowing them to be more robust and achieve better generalization. Bai et al. (2019) designed equilibrium models where the output $z$ was constrained to be a steady state, $z^* = f_\theta(z^*, x)$. Another example of implicit layer architectures is seen in Amos & Kolter (2017); Agrawal et al. (2019) set $z$ to be the solution of convex optimization problems.

Deep Implicit Models essentially removed the design bottleneck of choosing the "depth" of neural networks. Instead, these models use a "tolerance" to determine the accuracy to which the constraint needs to be satisfied. Additionally, many of these models only require $O(1)$ memory for backpropagation, thus alluding to potential increased efficiency over their explicit layer counterparts. However, evaluating these models require solving differential equations (Chen et al., 2018; Liu et al., 2019), non-linear equations (Bai et al., 2019), convex optimization problems (Amos & Kolter, 2017; Agrawal et al., 2019), etc. Numerous authors (Dupont et al., 2019; Grathwohl et al., 2018; Finlay et al., 2020; Kelly et al., 2020; Ghosh et al., 2020; Bai et al., 2021b) have noted that this solution process makes implicit models significantly slower in practice during training and prediction compared to explicit networks achieving similar accuracy.

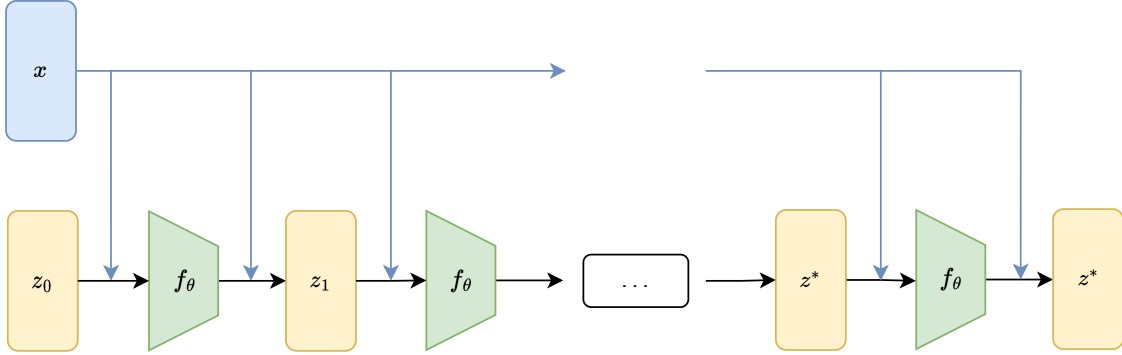

Figure 2: **Discrete DEQ Formulation**: Discrete DEQ Block where the input $x$ is injected at every iteration till the system (with initial state $z_0$) converges to a steady $z^*$. In Vanilla DEQ, $z_0 = 0$ while in Skip DEQ, an additional explicit model $g_\phi$ (which can potentially share the weights of $f_\theta$) is used to set the initial state $z_0 = g_\phi(x)$.

### 2.1 Neural Ordinary Differential Equations

Initial Value Problems (IVPs) are a class of ODEs that involve finding the state at a later time $t_1$, given the value $z_0$ at time $t_0$. Chen et al. (2018) proposed the Neural ODE framework, which uses neural networks to model the ODE dynamics

$$\frac{dz(t)}{dt} = f_\theta\left(z\right)$$

Using adaptive time stepping allows the model to operate at a variable continuous depth depending on the inputs. Removing the fixed depth constraint of Residual Networks provides a more expressive framework and offers several advantages in problems like density estimation Grathwohl et al. (2018), irregularly spaced time series problems Rubanova et al. (2019), etc. Training Neural ODEs using continuous adjoints has the added benefit of constant memory overhead. However, this benefit often leads to slower training since we need to backsolve an ODE. We defer the exact details of the continuous adjoint equations to Chen et al. (2018).

### 2.2 Deep Equilibrium Models

Deep Equilibrium Networks (DEQs) (Bai et al., 2019) are implicit models where the output space represents a steady-state solution. Intuitively, this represents infinitely deep neural networks with input injection, i.e., an infinite composition of explicit layers $z_{n+1} = f_\theta(z_n, x)$ with $z_0 = 0$ and $n \to \infty$. In practice, it is equivalent to evaluating a dynamical system until it reaches a steady state $z^* = f_\theta(z^*, x)$. Bai et al. (2019; 2020) perform nonlinear fixed point iterations of the discrete dynamical system using Broyden's method (Broyden, 1965; Bai et al., 2020) to reach this steady-state solution.

Evaluating DEQs requires solving a steady-state equation involving multiple evaluations of the explicit layer slowing down the forward pass. However, driving the solution to steady-state makes the backward pass very efficient (Johnson, 2006). Despite a potentially infinite number of evaluations of $f_\theta$ in the forward pass, backpropagation only requires solving a linear equation.

$$z^* = f_\theta(z^*, x)$$
$$\implies \frac{\partial z^*}{\partial \theta} = \frac{f_\theta(z^*, x)}{\partial z^*} \cdot \frac{\partial z^*}{\partial \theta} + \frac{\partial f_\theta(z^*, x)}{\partial \theta}$$
$$\implies \left(I - \frac{\partial f_\theta(z^*, x)}{\partial z^*}\right)\frac{\partial z^*}{\partial \theta} = \frac{\partial f_\theta(z^*, x)}{\partial \theta}$$

For backpropagation, we need the Vector-Jacobian Product (VJP):

$$\left(\frac{\partial z^*}{\partial \theta}\right)^T v = \left(\frac{\partial f_\theta(z^*, x)}{\partial \theta}\right)^T \left(I - \frac{\partial f_\theta(z^*, x)}{\partial z^*}\right)^{-T} v$$

$$= \left(\frac{\partial f_\theta(z^*, x)}{\partial \theta}\right)^T g$$

where $v$ is the gradients from layers after the DEQ module. Computing $\left(I - \frac{\partial f_\theta(z^*, x)}{\partial z^*}\right)^{-T}$ is expensive and makes DEQs non-scalable to high-dimensional problems. Instead, we solve the linear equation $g = \left(\frac{\partial f_\theta(z^*, x)}{\partial z^*}\right)^T g + v$ using Newton-Krylov Methods like GMRES (Saad & Schultz, 1986). To compute the final VJP, we need to compute $\left(\frac{\partial f_\theta(z^*, x)}{\partial \theta}\right)^T g$, which allows us to efficiently perform the backpropagation without explicitly computing the Jacobian.

### 2.2.1 Multiscale Deep Equilibrium Network

Multiscale modeling (Burt & Adelson, 1987) has been the central theme for several deep computer vision applications (Farabet et al., 2012; Yu & Koltun, 2015; Chen et al., 2016; 2017). The standard DEQ formulation drives a single feature vector to a steady state. Bai et al. (2020) proposed Multiscale DEQ (MDEQ) to learn coarse and fine-grained feature representations simultaneously. MDEQs operate at multiple feature scales $z = \{z_1, z_2, \ldots, z_n\}$, with the new equilibrium state $z^* = f_\theta(z_1^*, z_2^*, \ldots, z_n^*, x)$. All the feature vectors in an MDEQ are interdependent and are simultaneously driven to a steady state. Bai et al. (2020) used a Limited-Memory Broyden Solver (Broyden, 1965) to solve these large scale computer vision problems. We use this MDEQ formulation for all our classification experiments.

### 2.2.2 Jacobian Stabilization

Infinite composition of a function $f_\theta$ does not necessarily lead to a steady-state – chaos, periodicity, divergence, etc., are other possible asymptotic behaviors. The Jacobian Matrix $J_{f_\theta}(z^*)$ controls the existence of a stable steady-state and influences the convergence of DEQs in the forward and backward passes. Bai et al. (2021b) describes how controlling the spectral radius of $J_{f_\theta}(z^*)$ would prevent simpler iterative solvers from diverging or oscillating. Bai et al. (2021b) introduce a Jacobian term to the training objective to regularize the model training. The authors use the Hutchinson estimator (Hutchinson, 1989) to compute and regularize the Frobenius norm of the Jacobian.

$$\mathcal{L}_{jac} = \lambda_{jac} \frac{\|\epsilon^T J_{f_\theta}(z^*)\|_2^2}{d}; \quad \epsilon \sim \mathcal{N}(0, I_d)$$

While well-motivated, the disadvantage of this method is that the Hutchinson trace estimator requires automatic differentiation in the loss function, thus requiring higher order differentiation in the training process and greatly increasing the training costs. However, in return for the increased cost, it was demonstrated that increased robustness followed, along with faster forward passes in the trained results. Our methods are orthogonal to the Jacobian stabilization process. In Section 4, we provide empirical evidence on composing our models with Jacobian Stabilization to achieve even more robust results.

## 3 Methods

### 3.1 Continuous Deep Equilibrium Networks

Deep Equilibrium Models have traditionally been formulated as steady-state problems for a discrete dynamical system. However, discrete dynamical systems come with a variety of shortcomings. Consider the following linear discrete dynamical system (See Figure 3):

$$u_{n+1} = \alpha \cdot u_n \qquad \texttt{where } \|\alpha\| < 1 \texttt{ and } u_0 = 1$$

This system converges to a steady state of $u_\infty = 0$. However, in many cases, this convergence can be relatively slow. If $\alpha = 0.9$, then after 10 steps, the value is $u_{10} = 0.35$ because a small amount only reduces each successive step. Thus convergence could only be accelerated by taking many steps together. Even further, if $\alpha = -0.9$, the value ping-pongs over the steady state $u_1 = -0.9$, meaning that if we could take some fractional step $u_{\delta t}$ then it would be possible to approach the steady state much faster.

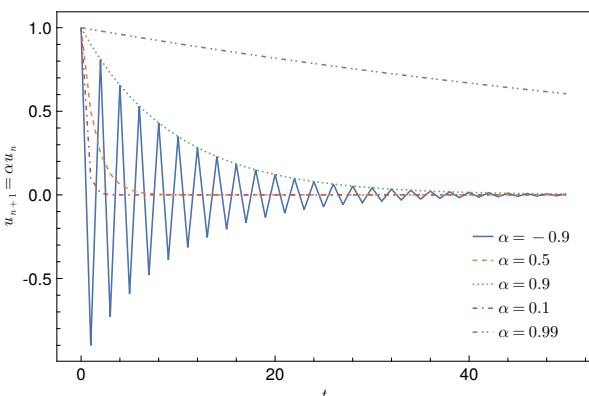

Figure 3: **Slow Convergence of Simple Linear Discrete Dynamical Systems**

Rico-Martinez et al. (1992); Bulsari (1995) describe several other shortcomings of using discrete steady-state dynamics over continuous steady-state dynamics. These issues combined motivate changing from a discrete description of the system (the fixed point or Broyden's method approach) to a continuous description of the system that allows adaptivity to change the stepping behavior and accelerate convergence.

To this end, we propose an alternate formulation for DEQs by modeling a continuous dynamical system (Continuous DEQ) where the forward pass is represented by an ODE which is solved from $t_0 = 0$ to $t_1 = \infty$:

$$\frac{dz}{dt} = f_\theta(z, x) - z$$

where $f_\theta$ is an explicit neural network. Continuous DEQs leverage fast adaptive ODE solvers, which terminate automatically once the solution is close to a steady state, i.e., $\frac{dz^*}{dt} = 0$, which then satisfies $f_\theta(z^*, x) = z^*$ and is thus the solution to the same implicit system as before.

The Continuous DEQ can be considered an infinite-time neural ODE in this form. However, almost paradoxically, the infinite time version is cheaper to train than the finite time version as its solution is the solution to the nonlinear system, meaning the same implicit differentiation formula of the original DEQ holds for the derivative. This means that no backpropagation through the steps is required for the Continuous DEQ, and only a linear system must be solved. In Section 4, we empirically demonstrate that Continuous DEQs outperform Neural ODEs in terms of training time while achieving similar accuracies.

### 3.2 Skip Deep Equilibrium Networks

Bai et al. (2019; 2020) set the initial condition $u_0 = 0$ while solving a DEQ. Assuming the existence of a steady state, the solvers will converge given enough iterations. However, each iteration is expensive, and a poor guess of the initial condition makes the convergence slower. To counteract these issues, we introduce an alternate architecture for DEQ (Skip DEQ), where we use an explicit model $g_\phi$ to predict the initial condition for the steady-state problem $u_0 = g_\phi(x)^2$. We jointly optimize for $\{\theta, \phi\}$ by adding an auxiliary loss function:

$$\mathcal{L}_{\texttt{skip}} = \lambda_{\texttt{skip}} \| f_\theta(z^*, x) - g_\phi(x) \|$$

---

[2] We note that the concurrent work Bai et al. (2021a) introduced a similar formulation as a part of HyperDEQ

| Model | Jacobian Reg. | # of Params | Test Accuracy (%) | Testing NFE | Training Time (min) | Prediction Time (s / batch) |
|---|---|---|---|---|---|---|
| Vanilla DEQ | ✗ | 138K | $97.926 \pm 0.107$ | $18.345 \pm 0.732$ | $5.197 \pm 1.106$ | $0.038 \pm 0.009$ |
| | ✓ | | $98.123 \pm 0.025$ | $5.034 \pm 0.059$ | $7.321 \pm 0.454$ | $0.011 \pm 0.005$ |
| Skip DEQ | ✗ | 151K | $97.759 \pm 0.080$ | $4.001 \pm 0.001$ | $1.711 \pm 0.202$ | $0.010 \pm 0.001$ |
| | ✓ | | $97.749 \pm 0.141$ | $4.001 \pm 0.000$ | $6.019 \pm 0.234$ | $0.012 \pm 0.001$ |
| Skip Reg. DEQ | ✗ | 138K | $97.973 \pm 0.134$ | $4.001 \pm 0.000$ | $1.295 \pm 0.222$ | $0.010 \pm 0.001$ |
| | ✓ | | $98.016 \pm 0.049$ | $4.001 \pm 0.000$ | $5.128 \pm 0.241$ | $0.012 \pm 0.000$ |

Table 1: **MNIST Classification with Fully Connected Layers**: Skip Reg. Continuous DEQ without Jacobian Regularization takes *4× less training time* and *speeds up prediction time by* 4× compared to Continuous DEQ. Continuous DEQ with Jacobian Regularization has a similar prediction time but takes *6× more training time* than Skip Reg. Continuous DEQ. Using Skip variants *speeds up training by 1.42 × −4×*.

Intuitively, our explicit model $g_\phi$ better predicts a value closer to the steady-state (over the training iterations), and hence we need to perform fewer iterations during the forward pass. Given that its prediction is relatively free compared to the cost of the DEQ, this technique could decrease the cost of the DEQ by reducing the total number of iterations required. However, this prediction-correction approach still uses the result of the DEQ for its final predictions and thus should achieve robustness properties equal.

### 3.2.1 Skip Regularized DEQ: Regularization Scheme without Extra Parameters

One of the primary benefits of DEQs is the low memory footprint of these models (See Section 2). Introducing an explicit model $g_\phi$ increases the memory requirements for training. To alleviate this problem, we propose a regularization term to minimize the L1 distance between the first prediction of $f_\theta$ and the steady-state solution:

$$\mathcal{L}_{\texttt{skip}} = \lambda_{\texttt{skip}} \| f_\theta(z^*, x) - f_\theta(0, x) \|$$

This technique follows the same principle as the Skip DEQ where the DEQ's internal neural network is now treated as the prediction model. We hypothesize that this introduces an inductive bias in the model to learn simpler training dynamics.

## 4 Experiments

In this section, we consider the effectiveness of our proposed methods – Continuous DEQs and Skip DEQs – on the training and prediction timings. We consider the following baselines:

1. Discrete DEQs with L-Broyden Solver.

2. Jacobian Regularization of DEQs.[3]

3. Multi-Scale Neural ODEs with Input Injection: A modified Continuous Multiscale DEQ without the steady state convergence constraint.

Our primary metrics are classification accuracy, the number of function evaluations (NFEs), total training time, time for the backward pass, and prediction time per batch. We showcase the performance of our methods on – MNIST (LeCun et al., 1998), CIFAR-10 (Krizhevsky et al., 2009), SVHN (Netzer et al., 2011), & ImageNet (Deng et al., 2009). We use perform our experiments in Julia (Bezanson et al., 2017) using Lux.jl (Pal, 2022) and DifferentialEquations.jl (Rackauckas & Nie, 2017; Rackauckas et al., 2018; 2020).

---

[3]We note that due to limitations of our Automatic Differentiation system, we cannot perform Jacobian Regularization for Convolutional Models. However, our preliminary analysis suggests that the Skip DEQ and Continuous DEQ approaches are fully composable with Jacobian Regularization and provide better performance compared to using only Jacobian Regularization (See Table 1).

| Model | Continuous | # of Params | Test Accuracy (%) | Training Time (s / batch) | Backward Pass (s / batch) | Prediction Time (s / batch) |
|---|---|---|---|---|---|---|
| Vanilla DEQ | ✗ | 163546 | $81.233 \pm 0.097$ | $0.651 \pm 0.009$ | $0.075 \pm 0.001$ | $0.282 \pm 0.005$ |
| | ✓ | | $80.807 \pm 0.631$ | $0.753 \pm 0.017$ | $0.261 \pm 0.010$ | $0.136 \pm 0.010$ |
| Skip DEQ | ✗ | 200122 | $82.013 \pm 0.306$ | $0.717 \pm 0.022$ | $0.115 \pm 0.004$ | $0.274 \pm 0.005$ |
| | ✓ | | $80.807 \pm 0.230$ | $0.806 \pm 0.010$ | $0.293 \pm 0.004$ | $0.154 \pm 0.002$ |
| Skip Reg. DEQ | ✗ | 163546 | $81.170 \pm 0.356$ | $0.709 \pm 0.005$ | $0.114 \pm 0.002$ | $0.283 \pm 0.007$ |
| | ✓ | | $82.513 \pm 0.177$ | $0.679 \pm 0.015$ | $0.143 \pm 0.017$ | $0.154 \pm 0.003$ |
| Neural ODE | ✓ | 163546 | $83.543 \pm 0.393$ | $1.608 \pm 0.026$ | $1.240 \pm 0.021$ | $0.207 \pm 0.006$ |

Table 2: **CIFAR10 Classification with Small Neural Network**: Skip Reg. Continuous DEQ achieves the *highest test accuracy among DEQs*. Continuous DEQs are faster than Neural ODEs during training by a factor of *2 × −2.36×*, with a speedup of *4.2 × −8.67×* in the backward pass. We also observe a prediction speed-up for Continuous DEQs of *1.77 × −2.07×* against Discrete DEQs and *1.34 × −1.52×* against Neural ODE.

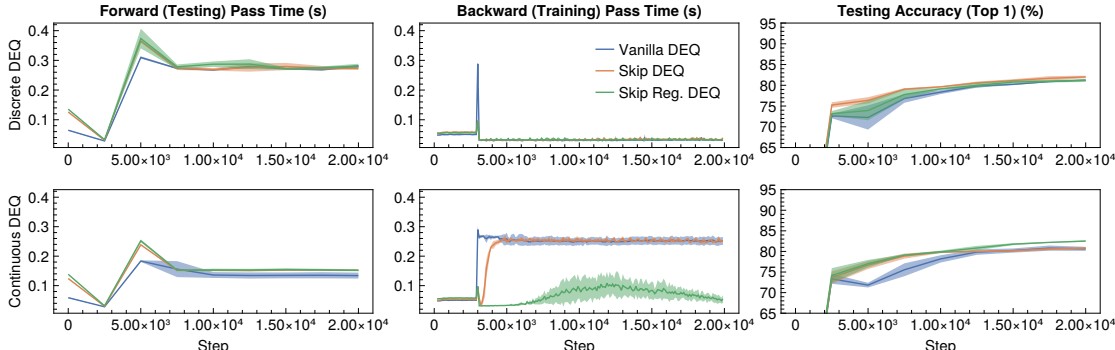

Figure 4: **CIFAR10 Classification with Small Neural Network**

## 4.1 MNIST Image Classification

**Training Details:** Following Kelly et al. (2020), our Fully Connected Model consists of 3 layers – a downsampling layer $\mathbb{R}^{784} \mapsto \mathbb{R}^{128}$, continuous DEQ layer $f_\theta : \mathbb{R}^{128} \mapsto \mathbb{R}^{128}$, and a linear classifier $\mathbb{R}^{128} \mapsto \mathbb{R}^{10}$.

For regularization, we use $\lambda_{\texttt{skip}} = 0.01$ and train the models for 25 epochs with a batch size of 32. We use Tsit5 (Tsitouras, 2011) with a relative tolerance for convergence of 0.005. For optimization, we use Adam (Kingma & Ba, 2014) with a constant learning rate of 0.001.

**Baselines:** We use continuous DEQ and continuous DEQ with Jacobian Stabilization as our baselines. We additionally compose Skip DEQs with Jacobian Stabilization in our benchmarks. For all experiments, we keep $\lambda_{\texttt{jac}} = 1.0$.

**Results:** We summarize our results in Table 1. Without Jacobian Stabilization, Skip Reg. Continuous DEQ has the highest testing accuracy of *97.973%* and has the *lowest training and prediction timings overall*. Using Jacobian Regularization, DEQ outperforms Skip DEQ models by *< 0.4%*, however, jacobian regularization increases training time by *1.4 − 4×*. Skip DEQ models can obtain the lowest prediction time per batch of *∼ 0.01s*.

## 4.2 CIFAR10 Image Classification

For all the baselines in this section, Vanilla DEQ is trained with the same training hyperparameters as the corresponding Skip DEQs (taken from Bai et al. (2020)). Multiscale Neural ODE with Input Injection is trained with the same hyperparameters as the corresponding Continuous DEQs.

| Model | Continuous | # of Params | Test Accuracy (%) | Training Time (s / batch) | Backward Pass (s / batch) | Prediction Time (s / batch) |
|---|---|---|---|---|---|---|
| Vanilla DEQ | ✗ | 10.63M | $88.913 \pm 0.287$ | $0.625 \pm 0.165$ | $0.111 \pm 0.021$ | $0.414 \pm 0.222$ |
|  | ✓ |  | $89.367 \pm 0.832$ | $1.284 \pm 0.011$ | $0.739 \pm 0.003$ | $0.606 \pm 0.010$ |
| Skip DEQ | ✗ | 11.19M | $88.783 \pm 0.178$ | $0.588 \pm 0.042$ | $0.112 \pm 0.006$ | $0.314 \pm 0.017$ |
|  | ✓ |  | $89.600 \pm 0.947$ | $0.697 \pm 0.012$ | $0.150 \pm 0.013$ | $0.625 \pm 0.004$ |
| Skip Reg. DEQ | ✗ | 10.63M | $88.773 \pm 0.115$ | $0.613 \pm 0.048$ | $0.109 \pm 0.008$ | $0.268 \pm 0.031$ |
|  | ✓ |  | $90.107 \pm 0.837$ | $0.660 \pm 0.019$ | $0.125 \pm 0.003$ | $0.634 \pm 0.019$ |
| Neural ODE | ✓ | 10.63M | $89.047 \pm 0.116$ | $5.267 \pm 0.078$ | $4.569 \pm 0.077$ | $0.573 \pm 0.010$ |

Table 3: **CIFAR10 Classification with Large Neural Network**: Skip Reg. Continuous DEQ achieves the *highest test accuracy*. Continuous DEQs are faster than Neural ODEs during training by a factor of $4.1 \times - 7.98\times$, with a speedup of $6.18 \times - 36.552\times$ in the backward pass. However, we observe a prediction slowdown for Continuous DEQs of $1.4 \times - 2.36\times$ against Discrete DEQs and $0.90 \times - 0.95\times$ against Neural ODE.

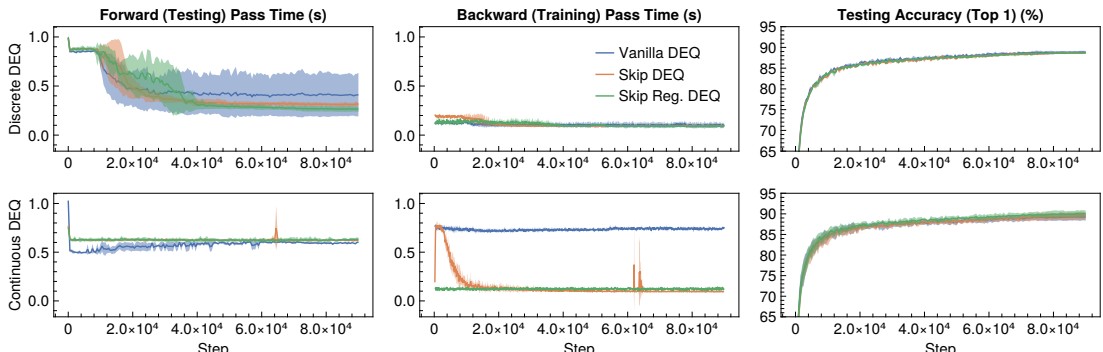

Figure 5: **CIFAR10 Classification with Large Neural Network**

### 4.2.1 Architecture with 200K parameters

**Training Details:** Our Multiscale DEQ architecture is the same as MDEQ-small architecture used in Bai et al. (2020). For the explicit network in Skip DEQ, we use the residual block and downsampling blocks from Bai et al. (2020) which account for the additional 58K trainable parameters.

We use a fixed regularization weight of $\lambda_{\mathtt{skip}} = 0.01$ and the models are trained for 20000 steps. We use a batch size of 128. For continuous models, we use VCAB3 (Wanner & Hairer, 1996) with a relative tolerance for convergence of 0.05. We use AdamW (Loshchilov & Hutter, 2017) optimizer with a cosine scheduling on the learning rate – starting from $10^{-3}$ and terminating at $10^{-6}$ – and a weight decay of $2.5 \times 10^{-6}$.

**Results:** We summarize our results in Table 2 and Figure 4. Continuous DEQs are faster than Neural ODEs during training by a factor of $2 \times - 2.36\times$, with a speedup of $4.2 \times - 8.67\times$ in the backward pass.

### 4.2.2 Architecture with 11M parameters

**Training Details:** Our Multiscale DEQ architecture is the same as MDEQ-large architecture used in Bai et al. (2020). For the explicit network in Skip DEQ, we use the residual block and downsampling blocks from Bai et al. (2020) which account for the additional 58K trainable parameters.

We use a fixed regularization weight of $\lambda_{\mathtt{skip}} = 0.01$ and the models are trained for 90000 steps. We use a batch size of 128. For continuous models, we use VCAB3 (Wanner & Hairer, 1996) with a relative tolerance for convergence of 0.05. We use Adam (Kingma & Ba, 2014) optimizer with a cosine scheduling on the learning rate – starting from $10^{-3}$ and terminating at $10^{-6}$.

| Model | Continuous | # of Params | Test Accuracy (Top 5) (%) | Training Time (s / batch) | Backward Pass (s / batch) | Prediction Time (s / batch) |
|---|---|---|---|---|---|---|
| Vanilla DEQ | ✗ | 17.91M | $81.809 \pm 0.115$ | $2.057 \pm 0.138$ | $0.195 \pm 0.007$ | $1.963 \pm 0.189$ |
| | ✓ | | $81.329 \pm 0.516$ | $3.131 \pm 0.027$ | $1.873 \pm 0.015$ | $1.506 \pm 0.027$ |
| Skip DEQ | ✗ | 18.47M | $81.717 \pm 0.452$ | $1.956 \pm 0.012$ | $0.194 \pm 0.001$ | $1.843 \pm 0.025$ |
| | ✓ | | $81.334 \pm 0.322$ | $2.016 \pm 0.129$ | $0.845 \pm 0.127$ | $1.575 \pm 0.053$ |
| Skip Reg. DEQ | ✗ | 17.91M | $81.611 \pm 0.369$ | $1.996 \pm 0.035$ | $0.539 \pm 0.023$ | $1.752 \pm 0.093$ |
| | ✓ | | $81.813 \pm 0.350$ | $1.607 \pm 0.044$ | $0.444 \pm 0.026$ | $1.560 \pm 0.021$ |

Table 4: **ImageNet Classification**: All the variants attain comparable evaluation accuracies. Skip (Reg.) accelerates the training of Continuous DEQ by $1.57 \times - 1.96\times$, with a reduction of $2.2 \times -4.2\times$ in the backward pass timings. However, we observe a marginal increase of $4\%$ in prediction timings for Skip (Reg.) Continuous DEQ compared against Continuous DEQ. For Discrete DEQs, Skip (Reg.) variants reduce the prediction timings by $6.5\% - 12\%$.

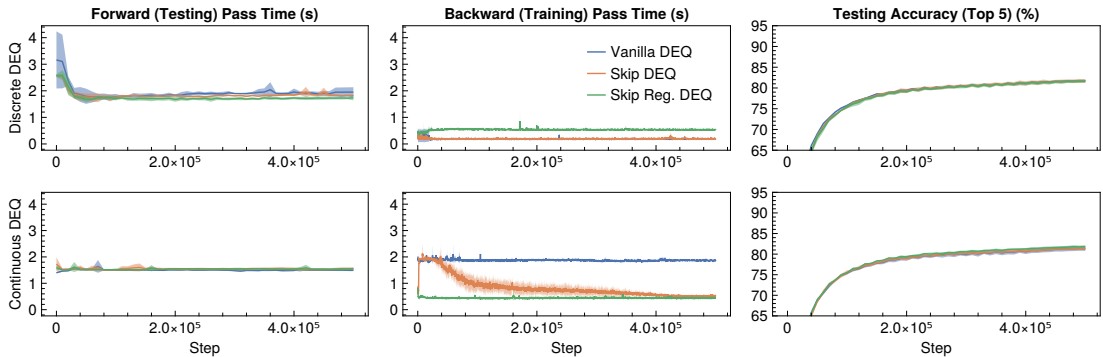

Figure 6: **ImageNet Classification**

**Results:** We summarize our results in Table 3 and Figure 5. Continuous DEQs are faster than Neural ODEs during training by a factor of $4.1 \times -7.98\times$, with a speedup of $6.18 \times -36.552\times$ in the backward pass.

### 4.3 ImageNet Image Classification

**Training Details:** Our Multiscale DEQ architecture is the same as MDEQ-small architecture used in Bai et al. (2020). For the explicit network in Skip DEQ, we use the residual block and downsampling blocks from Bai et al. (2020) which account for the additional 58K trainable parameters.

We use a fixed regularization weight of $\lambda_{\mathtt{skip}} = 0.01$, and the models are trained for 500000 steps. We use a batch size of 64. For continuous models, we use VCAB3 (Wanner & Hairer, 1996) with a relative tolerance for convergence of 0.05. We use SGD with a momentum of 0.9 and weight decay of $10^{-6}$. We use a step LR scheduling reducing the learning rate from 0.05 by a multiplicative factor of 0.1 at steps 100000, 150000, and 250000.

**Baselines:** Vanilla DEQ is trained with the same training hyperparameters as the corresponding Skip DEQs (taken from (Bai et al., 2020))[4].

**Results:** We summarize our results in Table 4 and Figure 6. Skip (Reg.) variants accelerate the training of Continuous DEQ by $1.57 \times -1.96\times$, with a reduction of $2.2 \times -4.2\times$ in the backward pass timings.

---

[4]When training MultiScale Neural ODE with the same configuration as Continuous DEQ, we observed a $8\times$ slower backward pass which made the training of the baseline infeasible.

## 5 Related Works

### 5.1 Implicit Models

Implicit Models have obtained competitive results in image processing (Bai et al., 2020), generative modeling (Grathwohl et al., 2018), time-series prediction (Rubanova et al., 2019), etc, at a fraction of memory requirements for explicit models. Additionally, Kawaguchi (2021) show that for a certain class of DEQs convergence to global optima is guaranteed at a linear rate. However, the slow training and prediction timings (Dupont et al., 2019; Kelly et al., 2020; Finlay et al., 2020; Ghosh et al., 2020; Pal et al., 2021; Bai et al., 2021b) often overshadow these benefits.

### 5.2 Accelerating Neural ODEs

Finlay et al. (2020); Kelly et al. (2020) used higher-order regularization terms to constrain the space of learnable dynamics for Neural ODEs. Despite speeding up predictions, these models often increase the training time by 7x (Pal et al., 2021). Alternatively, Ghosh et al. (2020) randomized the endpoint of Neural ODEs to incentivize simpler dynamics. Pal et al. (2021) used internal solver heuristics – local error and stiffness estimates – to control the learned dynamics in a way that decreased both prediction and training time. Xia et al. (2021) rewrite Neural ODEs as heavy ball ODEs to accelerate both forward and backward passes. Djeumou et al. (2022) replace ODE solvers in the forward with a Taylor-Lagrange expansion and report significantly better training and prediction times.

Regularized Neural ODEs can not be directly extended to discrete DEQs (Bai et al., 2019; 2020). Our continuous formulation introduces the potential to extend Xia et al. (2021); Djeumou et al. (2022) to DEQs. However, these methods benefit from the structure in the backward pass, which does not apply to DEQs. Additionally, relying on discrete sensitivity analysis (Pal et al., 2021) nullifies the benefit of a cost-effective backward pass.

### 5.3 Accelerating DEQs

Bai et al. (2021b) uses second-order derivatives to regularize the Jacobian, stabilizing the training and prediction timings of DEQs. Fung et al. (2022) proposes a Jacobian-Free Backpropagation Model, which accelerates solving the Linear Equation in the backward pass. Our work complements these models and can be freely composed with them. We have shown that a poor initial condition harms convergence, and a better estimate for the same leads to faster training and prediction. We hypothesize that combining these methods would lead to more stable and faster convergence, demonstrating this possibility with the Jacobian regularization Skip DEQ.

## 6 Discussion

We have empirically shown the effectiveness of Continuous DEQs as a faster alternative for Neural ODEs. Consistent with the ablation studies in Bai et al. (2021a), we see that Skip DEQ in itself doesn't significantly improve the prediction or training timings for Discrete DEQs. Skip Reg. DEQ does, however, speeds up the inference for larger Discrete DEQs. However, combining Skip DEQ and Skip Reg. DEQ with Continuous DEQs, enable a speedup in backward pass by over $2.8 - 5.9\times$. We hypothesize that this improvement is due to reduction in the condition number, which results in faster convergence of GMRES in the backward pass, however, acertaining this would require furthur investigation. We have demonstrated that our improvements to DEQs and Neural ODEs enable the drop-in replacement of Skip Continuous DEQs in any classical deep learning problem where continuous implicit models were previously employed.

### 6.1 Limitations

We observe the following limitations for our proposed methods:

- Reformulating a Neural ODE as a Continuous DEQ is valid, when the actual dynamics of the system doesn't matter. This holds true for all applications of Neural ODEs to classical Deep Learning problems.

- Continuous DEQs are slower than their Discrete counterparts for larger models (without any significant improvement to accuracy), hence the authors recommend their usage only for cases where a continuous model is truly needed.

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
