# OpenReview forum: "Continuous Deep Equilibrium Models: Training Neural Odes Faster by Integrating Them to Infinity"
_TMLR — Rejected by TMLR_

### Review · Reviewer_TVFN · 2023-04-10

**Summary Of Contributions:**

This paper studies continuous DEQs defined in section 3.1. Section 4 shows that they can result in a better training time, NFE, and accuracy than the discretized variants on. Tables 1/2 highlight the importance of the Skip DEQ and the regularization.

**Audience:**

Yes

**Broader Impact Concerns:**

I do not have any broader impact concerns

**Claims And Evidence:**

Yes

**Requested Changes:**

I do not have any at this point

**Strengths And Weaknesses:**

+ [+] This is a clearly-written paper that expresses the idea of a continuous-time DEQ, limitations of that formulation, and explore two improvements (skip/reg) for discrete and continuous DEQs.
+ [+] The results on CIFAR10/ImageNet seem consistent with previously published results on DEQS and seems experimentally sound.
+ [+/-] The continuous DEQ does not always have better performance than the baseline discrete DEQ (e.g. in Table 2 on CIFAR10 where the accuracies are slightly worse and in Table 4). While it is unfortunate there isn't a clear methodological improvement, some trends remain, such as the improvements of the skip/reg terms. These results seem scientifically honest about what was tried and interesting to understand how a continuous DEQ variant performs

---

> ### Author Response · Authors · 2023-05-12
>
> We would like to thank the reviewer for the review and that they found our paper to be “experimentally sound” and “clearly written.” We agree with the reviewer that it is unfortunate that continuous DEQs don’t outperform discrete DEQs. However, we still observe that continuous DEQs can be a drop-in replacement for Neural ODEs modeling time-invariant dynamics.

---

### Review · Reviewer_xXG1 · 2023-04-24

**Summary Of Contributions:**

The authors propose several  improvements to the original Deep Equilibrium Models architectures in order to make the model faster to train.

Notably the authors propose to add a:

1. an initial condition and a small displacement constraint: namely to predict the final $z^\star $ from the input $x$ and that the initial and final points must be close to one another.
2. a continuous formulation that finds the fixed point of the forward $f_\theta$ considering a dynamical system from $dz/dt =  f_\theta(z, x) - z$. Solving from 0 do $+\infty$ enables the authors to retrieve the fixed state of $z$.

Both contributions are validated experimentally on image classification tasks.

**Audience:**

Yes

**Broader Impact Concerns:**

This work addresses learning procedure for neural networks. As such it does not  require a Broader impact section.

**Claims And Evidence:**

Yes

**Requested Changes:**

(R1) While the proposed regularization (skip) are sounded, I highly recommend the inclusion of ablation studies to validate the relative influence of each of the model components to the performances.

(R2) The skip regularizations propose to both learn a starting point and to constraint $f_\theta$  to obtain only "small" deviation models.
In that perspective my questions are: what is the influence from integrating to infinity instead of a fixed (compared to a potentially small given the proposed constrained) bound ? What is the influence of the integration step on the convergence ?

(R3) While I concur with the authors that DEQ enables to abstract the number of layers of a NN, so does the NeuralODE framework. Therefore, I believe that without the constraints, the authors formulation as a DEQ is a modification of the forward pass (and the bound) of a NeuralODE. Thus, I strongly recommend the inclusion of theoretical considerations on the convergence of the integral, and the condition for solving a steady state equation. At least an extensive study on how to properly parameterize the NN so that the integral converges properly.

**Strengths And Weaknesses:**

Strength:

(S1) The writing of the paper is clear.
(S2) The presentation is easy to follow and the different ideas introduced in the paper flow interestingly in the authors presentation.
(S3) The set of experiments conducted in the main paper seem to validate the approach showing that the authors algorithm performs on par with some selected baselines.

Weaknesses:

(W1) The experiments are limited to image classification tasks. It would have been nice to expose the advantage of the method in various settings.
(W2) The computational performances are, in my opinion, under-analyzed. Indeed, it seem that the major source  of improvement of the computational cost comes from the introduced regularization.  Can the authors comment on that topic ?
(W3) There is also a limitation in the presentation of the work : First, despite being interesting the continuous time formulation propose to integrate on a non finite set to retrieve the steady state. Can the authors think of condition on $f_\theta$ where such integration scheme does not converge to the steady state. For instance, I can imagine that specific initialization / activation function can lead to bad normalization and diverging integrals ?
(W4) Despite being interesting and adapted to the DEQ formulation, the proposed regularization are not supported by either exhaustive experimental analysis or theoretically for instance analyzing which ODE is actually solved during training / inference.

Minor:
I suggest the authors to include the overall model loss in the main paper for clarity.

---

> ### Author Response · Authors · 2023-05-12
>
> We thank the reviewer for their review. We are glad that the reviewer found our writing to be “clear” and to “flow interestingly.” We think the following discussion should clarify the weaknesses pointed out by the reviewer.
>
> (W1) We restrict ourselves to image classification to allow a direct comparison to Multiscale Discrete DEQs. Though we agree with the reviewer that a detailed future study on additional tasks would be a valuable contribution.
>
> (W2) Our experiments suggest that for smaller problems using the regularization schemes gives us significant gains. However, for larger steady-state systems, the predictions by Skip (Reg) don’t really seem to help discrete DEQ.
>
> However, for Continuous DEQs, the regularization schemes accelerate the training by accelerating the **backward pass** (not the forward pass). Our leading hypothesis is that the regularization modifies the condition number of the linear system for the backward pass (for the large systems). However, verifying this is hard since we must construct the entire Jacobian matrix in the backward pass.
>
> (W3) Our experimental analysis suggested that tested methods for discrete DEQ initialization and training translate extremely well to continuous DEQs. All the experimental results reported used the same hyperparameters that were reported in the corresponding discrete DEQ papers. As an example, we have found that default explicit neural network initialization like Xavier initialization can lead to divergence; an easy solution is to perform a Gaussian Initialization with 0 mean and a small standard deviation of ~0.01. Additionally, pretraining by unrolling (like an RNN and not caring about the Steady State Problem) for the first few thousand iterations helps stabilize the dynamics. We left out these details for the sake of brevity, but we can add them to the main paper if these are valuable insights.
>
> (W4) We believe Tables 3 and 4 demonstrate that including the proposed regularization schemes for Continuous DEQs always improves the training performance. However, we note that the performance gain doesn’t come from the forward pass (see prediction timings), and hence it comes entirely from the improved backward pass. This shows that the linear solve in the backward pass is being accelerated. Since we are using GMRES for the backward pass and a modified condition number is what can speed up the convergence of the iterative linear solver, our leading hypothesis is that the regularization modifies the dynamical system to have a faster backward pass (which we agree is a surprising discovery but is something that definitely holds up from our experimental data).

---

### Review · Reviewer_SkHr · 2023-04-28

**Summary Of Contributions:**

The reviewed work proposes modifications to the popular Deep Equilibrium Models (DEQ) of Bai et al.
It suggests casting the discrete fixed point iteration as a continuous ODE, akin to the popular neural ODE model. To accelerate the training of the resulting model, it suggests using a second neural network to learn a good initial guess.
This methodology is then put to test on image classification benchmarks (MNIST, CIFAR10, ImageNET).

**Audience:**

No

**Broader Impact Concerns:**

No concerns

**Claims And Evidence:**

No

**Requested Changes:**

As noted above I am not convinced by the improvements of the proposed methodology. The paper would have to be substantially rewritten/rescoped to not be confusing to the community.

**Strengths And Weaknesses:**

Strengths:

By using more potent ODE solvers, passing from to continuous steady-state equations may indeed yield benefits in the future.

The skip DEQ variant appears to sometimes yield modest improvements over DEQ.

Weaknesses:

The main novelty proposed in the paper (and advertised in abstract and title) of passing to continuous time, does not appear to result in any significant improvement over the most standard baseline (discrete DEQ without further modifications). The inclusion of Neural ODEs appears largely as a strawman, given that discrete DEQ already exist and achieve significantly better results on the benchmarks under consideration. The results on skip regularization and observation that DEQs are a formidable competitor to Neural ODEs may be of some interest, but to move the community forward the paper needs to be, at least, thoroughly rewritten and rescoped.

---

> ### Author Response · Authors · 2023-05-12
>
> We thank the reviewer for the review. We agree with the reviewer that it is unfortunate that continuous DEQs don’t offer computational benefits over Discrete DEQs. However, this is not entirely surprising since solving ODEs with an infinite time span is slower than nonlinear problems. However, as discussed in https://docs.sciml.ai/NonlinearSolve/stable/solvers/SteadyStateSolvers/, solving Nonlinear Problems involving replacing the time-dependency to $\infty$ and hence solving an ODEProblem is the only way to get the correct unique fixed point. This and other benefits highlighted in the citations in the main text motivate continuous DEQs.
>
> We strongly disagree with the reviewer about “The inclusion of Neural ODEs appears largely as a strawman”: Discrete DEQs are indeed competitive to Neural ODEs. However, they model completely different dynamical systems, and hence they cannot be readily used as a drop-in replacement for Neural ODEs. However, for time-invariant Neural ODEs, Continuous DEQs are not only a drop-in replacement but also significantly faster to train, as evidenced by our experimental results.
>
> We believe this should clarify the questions posed by the reviewer. We are additionally surprised that even though the reviewer found that our paper might be of interest to the Neural ODE community, they recommend that this paper doesn’t fit with the TMLR audience. Additionally, no clear argument is made against the presented evidence for the claims (mentioned in the paper) with another *no* recommendation.

---

### Review · Reviewer_xm75 · 2023-05-01

**Summary Of Contributions:**

This paper proposes a continuous time variant of deep equilibrium model, which is claimed to perform faster than its discrete time version. In addition, it proposes a skip DEQ architecture aiming to find a better initial condition for solving DEQ so that the DEQ is solved in fewer iterations.

**Audience:**

No

**Claims And Evidence:**

No

**Requested Changes:**

see the comments in the "weakness" section.

**Strengths And Weaknesses:**

This paper is ambiguous on the proposed methods, and lacks description of the design and some important justifications.

> For continuous time DEQ, the paper fails to describe the solver of the continuous DEQ, while claiming the continuous DEQ is solved faster. A detailed description of the solver and a justification how the solver is cheaper to train are necessary. In addition, the following claims should also be justified: “the infinite time version is cheaper …  as its solution is the solution to the nonlinear system”, and “no backpropagation … is required … , and only a linear system must be solved”.

> For the skip DEQ, the paper did not explain the following important aspects:

>> 1,  how the initial condition is predicted (for example, what is the explicit model g?, why does the f(z^*) is known in the loss function?, how is g trained?) All the essential pieces are missing.

>> 2, why does the model g give better predictions on the initial condition? This is totally not explained in the paper.

>> 3, I could not understand why “assuming the existence of a steady state, the solvers will converge …”. In principle, the existence of a steady state is not enough to guarantee the solver converges to it. The authors should explain.

The paper uses a counter-example to illustrate the issue with discrete time DEQ (see figure 3). However, it is not explained whether the proposed continuous time version can solve this issue.

The background section (section 2) is not well written, and the background and problem setup remain unclear. This section seems to focus on related works and is filled with short comments on them. However, the problem is never clearly set up.

Minor:

in the title: Odes → ODEs

---

> ### Author Response · Authors · 2023-05-12
>
> We thank the reviewer for their review. We would clarify some of the questions raised in this rebuttal. We note that to keep the paper concise and to avoid repeating information widely known in the implicit machine learning community, some of these details were left out and are available via our (to be) open source (d) implementation:
>
> 1. “A detailed description of the solver and a justification how the solver is cheaper to train are necessary”
>     * Any ODE Solver can be used for this. Discussion of ODE solvers is beyond the scope of this paper. For the experiments, we either use Tsit5 (an RK4 scheme) or 3rd-order Adam Bashforth (a multistep method).
>     * We do not claim that ODE solvers are “cheaper” than Nonlinear Solvers in the paper. It is almost certainly the other way around. The training is, however, “cheaper” as described in the next step.
> 2. “the infinite time version is cheaper … as its solution is the solution to the nonlinear system”, and “no backpropagation … is required … , and only a linear system must be solved”
>     * A finite-time neural ODE has to be trained using either continuous or discrete sensitivity analysis. Discrete Sensitivity analysis requires backpropagating through the solver, which often involves hundreds of calls for Vector-Jacobian Products for the neural network. For continuous sensitivity, we must backsolve the neural ODE, which again requires hundreds of calls to the Neural Network.
>     * It is well known in practice that solving a **linear equation** is cheaper than solving a **nonlinear ODE system** (for the finite time Neural ODE version). Additionally, *all our experimental results comparing Neural ODE with Continuous DEQs are a direct testament to the fact that our claim is well justified*.
> 3. “how the initial condition is predicted (for example, what is the explicit model g?, why does the f(z^*) is known in the loss function?, how is g trained?)” -- For our experiments, we fixed $g$ to be a copy of $f$ with minor modifications that are available in the code linked in the paper. $f(z^*)$ is known since we are solving the steady state problem, which gives us $f(z^*)$. The loss function involving $g$ is completely differentiable and hence can be trained using gradient descent or any other gradient-based training algorithms.
> 4. “why does the model g give better predictions on the initial condition?” -- $g$ attempts to bias the initial condition for the ODEProblem/NonlinearProblem to be close to the steady state, which should lead to faster convergence.
> 5. “I could not understand why “assuming the existence of a steady state, the solvers will converge …”. In principle, the existence of a steady state is not enough to guarantee the solver converges to it. The authors should explain.” -- This is clearly a mistake in writing on our part. We will update the text accordingly.
> 6. “it is not explained whether the proposed continuous time version can solve this issue”: The citations in the main text “Rico-Martinez et al. (1992); Bulsari (1995)” discuss this exact issue.
>
> We believe these details should clarify the questions posed by the reviewer. However, we reiterate that most of these details are already widely known, as evidenced by none of the other reviewers raising similar concerns. Instead, the other reviewers have praised our writing for being “clearly-written,” “flows interestingly,” and are contributions to be “validated experimentally.”
>
> Additionally, it is unclear what the reviewer expects in the background section since it clearly setups what Neural ODEs and Deep Equilibrium Networks are. However, we are willing to update or clarify the writing if any actionable review is presented regarding that.

---

### Review · Reviewer_nfvd · 2023-06-01

**Summary Of Contributions:**

Apologies for the late review.

The authors empirically study the following settings for neural ODEs on four datasets:
1. Skip-DEQ, which uses a small explicit neural network to guess the solution to the neural ODE and use this as an initial solution,
2. An extension of skip-DEQ, which additionally encourages the first iterate to be close to the steady state solution,
3. Solving the neural ODE at $t_1 = \infty$.


**Audience:**

Yes

**Broader Impact Concerns:**

None.

**Claims And Evidence:**

Yes

**Requested Changes:**

See weaknesses above. Also the following questions and minor points:

Questions:
- Section 3.2.1. Did you also try doing this for arbitrary future iterations? E.g. Could do this with iteration $1$, $2$, ..., $n$. If not, would you expect this to be helpful --- why or why not?
- Continuous time DEQs are faster than Neural ODEs, but continuous time DEQs appear to be slower than discrete time DEQs, in some metrics quite considerably.

Minor:
- Item 4 of "Our main contributions include" does not make sense grammatically. "Our main contributions include we demonstrate the..."
- ``layers'' in the first sentecne in section 2 uses the wrong quotation marks. Further examples throughout.

**Strengths And Weaknesses:**

Strengths:
- The authors introduce a nice initialisation method for the forward pass of the solver, by using a neural network for the initial guess (section 3.2). Unfortunately, as the authors note, this has already been done in concurrent work (Bai et al. 2021). Since considerable time has passed since this other work, I wonder whether it is still fair to claim this as a contribution, or rather just reference the existing approach. It is possible that the author's contribution is already available in a preprint form with an earlier timestamp (I did not check), in which case it is probably okay to still claim this as a contribution.


Weaknesses:
- Somewhere behind section 2.2.2, it would be nice to hint at the mathematics behind the various claims involving stability, infinitely many compositions, convergence etc. alluded to in the previous parts of the text. For example, the second sentence of section 2.2 is true if $f_\theta(\cdot, x)$ is a contraction mapping, but not necessarily otherwise. This is the motivation behind controlling the spectral norm of the Jacobian matrix. Mentioning the contraction principle and/or Banach's fixed point theorem would greatly assist in framing the problem and scope. It is still unclear at this point whether we care about ``ill-posed'' DEQs that are sometimes used in practice, such as those that admit multiple fixed points. Previously you alluded to convex optimisation problems, which may also admit multiple minima.
- Last part of section 2.2.2. What does ``increased robustness'' mean?
- Agree that even stable discrete systems can take a long time to converge if the Jacobian norm of the mapping is close to one. The same also applies to stable continuous time systems. That's why we have accelerated solvers (for both discrete and continuous), that do not involve simply simulating the dynamics. Discrete and continuous time systems can both be solved without simulating the dynamics, so the motivation for why a continuous time system over a discrete time system is not clear at this point in the manuscript. I.e. it is not clear why you say ``To this end...''
- I do not understand the fundamental difference between the equation above section 3.2 and the equation below section 2.1. Can you elaborate further? Is it just that $t_1$ is replaced by $\infty$?
- Section 3.2. The following claim is incorrect ``Assuming the existence of a steady state, the solvers will converge given enough iterations''. There are many examples of functions that admit a (possibly unique) steady state, where solvers fail to converge to the steady state.
- I am a bit confused about the message of this paper. It seems like it is really about solving neural ODEs with $t_1 = \infty$, and has little to do with DEQs (apart from the high level conceptual background). Continuous time systems are considerably slower than DEQs, however, if the authors framed the paper instead about neural ODEs, the advantages would be more clear. However, even with these advantages, it seems as though the gains in performance are very marginal, there is no theory in the paper justifying the idea, and the idea of setting $t_1 = \infty$ does not have enough meat in it to warrant a journal publication without more extensive empirical evaluation.

My understanding is that the paper has the following contributions:
1. Skip-DEQ initialisation for the forward pass (which was already introduced in 2021 by Bai et al.)
2. A minor extension of Skip-DEQ that also encourages the first iterate to be close to the solution, with limited discussion.
3. Replacing $t_1=\infty$ in a neural ODE
4. Some non-conclusive experiments about the benefits of 1,2 and 3.

My feeling is that these contributions are not novel enough to be of interest to the TMLR community, because of their incremental nature. Furthermore, little to no conceptual or theoretical justification is given for any of the proposed practical measures. There are very minor issues around correctness which should be addressed, but the claims of the authors seem to be mostly correct. If my understanding of the paper is not complete, and the authors point this out, I am happy to consider reassessing.

---

> ### Comment · Reviewer_nfvd · 2023-06-04
> **Comment**
>
> After reading the other reviews, I was surprised to see that 2 reviewers indicated "Yes" to the audience criteria. This means that at least some individuals in TMLR's audience are interested in knowing the findings of this paper. I therefore increase my score to "Yes" for this criteria. For the claims and evidence criteria, while the manuscript does contain some minor issues around claims, most of these do not seem to be central to the story of the paper and could be fixed with some small edits.
>
> My recommendation for this paper is extremely marginal. Initially I was voting for reject. While I personally do not find the paper to be of interest, I acknowledge that at least some of TMLR's audience are interested in the findings of the paper (as evidenced by the first two positive reviews). I therefore raise my assessment to an extremely marginal accept.

---

### Decision · Action_Editors · 2023-06-09

**Recommendation:** Reject

**Comment:**

Thank you for considering TMLR. The paper should be rewritten to clarify better its contribution, evidence for main claims, and relation to prior work, before being ready for publication. The paper has valuable findings such as the ablations, but they are presented in a highly unclear manner.

A major revision of the paper is welcomed for resubmission.

**Audience:**

To emphasize key points raised by reviewers: Reviewer TVFN wrote that findings around ablations (the third contribution) might be of interest to the community; Reviewer SkHr notes on the other hand that the paper is unclear on the fact that the continuous formulation of DEQ generally does not improve performance compared to vanilla DEQ (the first main claim).

Less crucially, the background section is unclear. For example, it is difficult to understand what specifically is meant by “scaling in the backward pass” in the sentence “*We marry the idea of faster backward pass for DEQs and continuous modeling from Neural ODEs to create Infinite Time Neural ODEs which scale significantly better in the backward pass and drastically reduce the training time*.” in the introduction. Relatedly reviewers nfvd, SkHr, xXG1, and xm75 have raised issues regarding the presentation. Reviewer xm75 specifically points out that the background section does not clearly set up the problem and is primarily focused on related works.

It is important to note that two reviewers found paper writing clear, somewhat in disagreement with other reviewers.

In my opinion, the paper has to be substantially rewritten to be of sufficient interest and value to the community. Most importantly: (a) clarification on claims around continuous DEQ should be made, (b) the relation to prior work should be clarified.

**Claims And Evidence:**

The main contributions of the paper are (1) the introduction of continuous formulation of DEQ, (2) the introduction of Skip DEQ, and (3) ablation studies concerning the benefits of 1 and 2. The main claim is that (1) continuous formulation and (2) skip DEQ improve the performance of deep equilibrium models.

Two reviewers voted to reject, two voted to accept (one emphasizing that it is a very marginal recommendation), and one didn’t submit an official recommendation but can be assumed to vote to accept.

Reviewer SkHr key criticism is that the paper is unclear on the fact that the continuous formulation of DEQ generally does not improve performance compared to vanilla DEQ. The evidence for the first main claim is in my opinion insufficient, and I agree with Reviewer SkHr.

While TMLR is not focused on novelty, it is important to note that contribution (2) has been already introduced in 2021 in Bai et al, and the relation to prior work should be clarified. Currently, it is (in my opinion) not sufficiently surfaced by being only disclosed in a footnote “We note that the concurrent work Bai et al. (2021a) introduced a similar formulation as a part of HyperDEQ”.

The paper should be rewritten to clarify better its contribution, evidence for main claims, and relation to prior work, before being ready for publication.

**Resubmission Of Major Revision:**

The authors may consider submitting a major revision at a later time.